# From Mosquito Ovaries to Ecdysone; from Ecdysone to *Wolbachia*: One Woman’s Career in Insect Biology

**DOI:** 10.3390/insects13080756

**Published:** 2022-08-22

**Authors:** Ann M. Fallon

**Affiliations:** Department of Entomology, University of Minnesota, 1980 Folwell Ave., St Paul, MN 55108, USA; fallo002@umn.edu

**Keywords:** cell cycle, cell line, flow cytometry, insect reproduction, mosquito, obligate intracellular bacterium, organ culture

## Abstract

**Simple Summary:**

My career in science, and specifically in insect biology, benefitted from post-war efforts to improve science education in the 1950s and 1960s and increased opportunities for women students and faculty; teachers, mentors, associates, colleagues and friends of all sorts helped along the way. Here, I recount the highlights of my research, focusing on studies involving the insect steroid hormone ecdysone, which regulates molting in insect larvae, and reproduction in adults. Ecdysone was the unknown “ovarian factor” involved in egg production in *Aedes aegypti* mosquitoes. In addition, ecdysone affects the process of cell division in cultured mosquito cells, causing a cell cycle arrest that enhances growth of an obligate symbiotic bacterium, *Wolbachia pipientis*. Since the mid-1920s, *Wolbachia* was known to infect ovaries of *Culex pipiens* mosquitoes. After *Wolbachia* sequences were discovered in DNA extracted from *Drosophila*, this obligate intracellular alpha proteobacterium was found to be widespread among insect species, and its effects on reproduction have important applications for control of insect pests. Most recently, effects of ecdysone on *Wolbachia* replication in mosquito cell lines led to insights that enhance production of infectious bacteria at levels suitable for microinjection into eggs and eventual genetic manipulation.

**Abstract:**

In anautogenous mosquitoes, synchronous development of terminal ovarian follicles after a blood meal provides an important model for studies on insect reproduction. Removal and implantation of ovaries, in vitro culture of dissected tissues and immunological assays for vitellogenin synthesis by the fat body showed that the *Aedes aegypti* (L.) (Diptera, Culicidae) mosquito ovary produces a factor essential for egg production. The discovery that the ovarian factor was the insect steroid hormone, ecdysone, provided a model for co-option of the larval hormones as reproductive hormones in adult insects. In later work on cultured mosquito cells, ecdysone was shown to arrest the cell cycle, resulting in an accumulation of diploid cells in G1, prior to initiation of DNA synthesis. Some mosquito species, such as *Culex pipiens* L. (Diptera, Culicidae), harbor the obligate intracellular bacterium, *Wolbachia pipientis* Hertig (Rickettsiales, Anaplasmataceae), in their reproductive tissues. When maintained in mosquito cell lines, *Wolbachia* abundance increases in ecdysone-arrested cells. This observation facilitated the recovery of high levels of *Wolbachia* from cultured cells for microinjection and genetic manipulation. In female *Culex pipiens*, it will be of interest to explore how hormonal cues that support initiation and progression of the vitellogenic cycle influence *Wolbachia* replication and transmission to subsequent generations via infected eggs.

## 1. Introduction

It started with asparagus beetles which my grandmother called lady bugs, and Japanese beetles, collected by the handful and drowned in kerosine. Then, there were the American Copper butterflies, which my dad told me would grow into Monarchs. A sphinx moth on a rhubarb leaf, which became two moths the next day, led me to conclude that new moths grow from the tail of older moths. By the time I was five, I had neighbors, and it was the grandfather of a childhood friend who inspired me to spend my life studying insects. I never spoke to him, but through his grandson I was introduced to a more rigorous version of natural history and learned that butterflies come from caterpillars, not from smaller butterflies. My mother helped me make a net from a coat-hanger and old lace curtains. By the time I started high school, I was preoccupied with rearing Saturniid moths, and noticed that from about 200 cocoons, male emergence peaked before that of females (Figure 1) and that mating was more likely to succeed when the male was older than the female. I wanted to figure out how much leaf mass was needed to make a moth, and tried to make a balance to weigh caterpillars against staples, thinking that I could weigh a staple sometime in the distant future. No luck there!

High school started out well—the first assignment in Biology was to bring in a caterpillar. I had a bit of concern over that, as high school was in a city 20 miles from home, with streets and sidewalks, and how would the teacher find the leaves to feed whatever the students collected? My solution was to contribute a *Hyalophora cecropia* Linnaeus (Lepidoptera, Saturniidae) larva that had finished feeding and was ready to spin, saving the teacher the problem of wild cherry leaves. I was too shy to ask to see the caterpillars collected by my classmates, but the teacher was impressed with the *cecropia* and that I reared them from eggs. To my dismay, however, she was unfamiliar with the word “instar.” My first term paper was on insecticides, drawing from Rachel Carson’s Silent Spring, my concerns about dad’s insecticides drifting to wild cherry trees and those toxic apples I did not eat. Maybe college…

At the University of Connecticut, I was fortunate to take an undergraduate course with Heinz Hermann. The textbook was “Cell Structure and Function” by Loewy and Siekevitz. When I handed in my final exam, Hermann asked me about my interests and introduced me to Hans Laufer, whose lab was investigating the effects of insect hormones on chromosome puffing in *Chironomus* salivary glands. I was amazed that undergraduates could work in a lab, and that it was possible to study insects from the inside. I best remember Ulrich Clever’s papers [1,2] describing the sequence of ecdysone-induced early and late puffs, and the concept of using inhibitors such as actinomycin D and cycloheximide to dissect pathways of gene activation. Memorizing banding patterns of polytene chromosomes and estimating puff sizes took up all the time I did not need for study, but eventually came the statistics, which suggested I did not have the “right” answer! More puffing patterns... how does one know when to stop?

Finding microscopy too subjective, I was attracted to the mysteries of Ouchterlony plates, which preoccupied postdoc Henry Hagedorn in his efforts to produce a figure for publication of his doctoral work with *Aedes aegypti* yolk protein. At that time, microscopic evidence suggested that mosquito vitellogenin was produced by the midgut [3], but Hagedorn suspected that as in moths, mosquito vitellogenin was produced by the fat body [4]. Hagedorn introduced me to alternative, biochemical approaches to study insects that could be quantified using radioactivity. One day, Hagedorn presented me with a set of papers and explained that it was time to choose a graduate program.

I had imagined that one had to be independently wealthy to be a scientist, and for the first time envisioned the possibility of a lifetime of study. As a grad student with Jerry Wyatt, I investigated guanosine 3′,5′-cyclic monophosphate (cyclic GMP) in the male accessory glands of crickets [5]; as a postdoc in the lab of Masayasu Nomura, I gained molecular experience by investigating post-translational regulation of ribosomal protein synthesis in *E. coli* [6]. In those days, individual labs purified their own restriction enzymes, and agarose gels were performed on raised glass plates covered with melted Vaseline to keep the agarose hydrated. Nomura’s group had recovered ribosomal protein genes on lambda transducing phages; today I work on *Wolbachia* phages [7], which may become a tool for genetic manipulation of a microorganism that infects an estimated half of all insect species.

I returned to insects by way of an additional postdoc with Victor Stollar (University of Medicine and Dentistry of New Jersey (UMDNJ)), whose group pioneered the development of somatic cell genetics with mosquito cell lines in the context of his work on arboviruses. Eventually, I accepted a faculty position at the newly established UMDNJ School of Osteopathic Medicine, whose dean “needed a woman for medical school admissions”. For the next five years, I was the only woman on that committee. Setting up a lab, chasing grants, training students and teaching left little time for my own experiments. Largely through students who sought training in molecular biology, my lab focused on the somatic cell genetics of mosquito cells in culture, emphasizing transfection and inducible gene expression [8,9,10,11,12]. 

Over the years, my students, postdocs and technical staff cloned and characterized mosquito ribosomal RNA and ribosomal protein genes [13,14,15,16,17] and explored aspects of their expression in the context of mosquito vitellogenesis [18,19,20,21] and their applications in phylogenetic analyses [22]. With cultured cells, I began to explore aspects of nucleic acid and protein metabolism [23,24,25,26,27,28,29,30], including characterization of genes encoding antimicrobial peptides [31,32,33]. An assigned lecture on the cell cycle and cancer informed later research with long-term associate Anna Gerenday, who used hydroxyurea to synchronize the mosquito cell cycle [34] and found a link between ecdysone and the cell cycle inhibitor protein, Dacapo [35]. With the publication of a *Wolbachia* genome [36] and the recognition that *Wolbachia* could be maintained in mosquito cells [37], I began my present efforts to optimize production of infectious *Wolbachia* in cell lines and to explore options for genetic manipulation of this obligate intracellular bacterium that causes potentially useful reproductive distortions in a wide range of arthropods. 

This paper will review in vitro experimental approaches and conceptual advances that have informed my work over several decades and remain particularly relevant to insect science today. These will include organ and cell culture, cell cycle and flow cytometry, as well as an introduction of *Wolbachia* into naive mosquito cells. Finally, I will describe new results that increase the yield of *Wolbachia* strain *w*AlbB by infecting mosquito cell feeder layers produced by treating cells with mitomycin C.

## 2. In Vitro Approaches to Insect Biology

One of the high points of my undergraduate work was the participation in studies showing that the insect steroid hormone, ecdysone, stimulates vitellogenin synthesis in the adult female mosquito fat body [38,39]. In earlier studies, Hagedorn and Judson [4] had demonstrated that the yolk protein precursor, called vitellogenin, was produced by mosquito fat body, rather than by the midgut. These studies underscored the similarity between moths and mosquitoes: in both, the fat body synthesizes a protein that passes through the hemolymph into developing eggs. In Saturniid moths, then popular for studies on insect reproduction due to their large size, ovariectomy removes the tissue that accumulates yolk, and vitellogenin protein, having nowhere to go, accumulates in the hemolymph [40]. To our surprise, however, no vitellogenin was produced by the fat body of ovariectomized mosquitoes. Were the ovary and fat body talking to each other through a hormonal signal?

As early as 1967, Larsen noted positive interactions resulting from the co-culture of dissected insect tissues, particularly in the presence of prothoracic glands [41]. Using a similar approach, we co-cultured mosquito ovaries and fat body in a medium that facilitated the incorporation of a radioactive amino acid into vitellogenin protein. Only newly synthesized proteins would contain radioactivity, which was quantified by precipitation with an antibody produced by injecting rabbits with an extract from mosquito eggs. Vitellogenin was produced when fat body was co-cultured with ovaries from blood-fed mosquitoes, suggesting that an unknown factor from the dissected ovary stimulated the fat body. Identification of the ovarian factor was guided by a report that in the absence of a blood meal, ovarian development in female *Ae. aegypti* was stimulated by injection of ecdysone [42] and by our own observations that ecdysone stimulated the synthesis of vitellogenin by fat body of unfed mosquitoes both in vivo and in vitro [43]. Drawing upon related circumstantial observations consistent with a possible role for ecdysone in other adult female insects, Hagedorn and coworkers used biochemical techniques to show that the *Ae. aegypti* ovarian factor was ecdysone [39].

The discovery that the ovary is the source of a reproductive hormone was facilitated by specific aspects of mosquito biology. First, the blood meal acts as a discrete trigger that initiates vitellogenin synthesis. Eggs are produced in batches, in which only the terminal ovarian follicles (roughly 100 per ovary) develop synchronously, while penultimate follicles remain quiescent. In contrast, adult Saturniid moths do not feed, are short-lived and their ovarioles consist of a chain of follicles in which development follows a gradient pattern. Although the distal follicle matures first, proximal follicles also accumulate yolk, each to a greater extent than the preceding follicle [44]. Parallel studies in diverse insects showed that like ecdysone, the juvenile hormone, known for its role in larval development, also participates in egg development in adult insects. Details of the precise roles of these two key hormones, and the ease of investigating their interactions, vary considerably and are influenced by life history traits such as food source and feeding patterns, longevity, diapause and migration [45]. This diversity has been particularly well-investigated in several Lepidoptera [44], while in Diptera, the most detailed studies are based on the yellow fever mosquito *Ae. aegypti* [46] and *Drosophila melanogaster* Meigen (Diptera, Drosophilidae) [47,48].

## 3. Insect Cell Culture

The conditions that supported the viability of dissected insect organs informed and complemented the efforts to develop permanent cell lines. While the earliest attempts to bridge the gap between the short-term maintenance of tissues and the establishment of continuous cell lines were discouraging [49,50], rapid advances accompanied systematic development of culture media based on insect hemolymph composition and recognition that deleterious effects of hemolymph phenoloxidase could be eliminated by heat treatment or inhibited with phenylthiourea [51,52]. When care is taken to sterilize the surface of source insects and avoid bacterial contamination by including antibiotics in the culture medium, dissected/disrupted tissues could remain functional, but not necessarily mitotic for hours to days. Over time, occasional outgrowths from cultured tissues included cells that continued to divide in vitro. The first permanent cell lines from insects were described in 1962 [53], and ancestors of many of the mosquito cell lines in use today, including the C7-10 cell line from *Aedes albopictus* (Skuse) (Diptera, Culicidae) used in my lab [54,55], date back to the late 1960s, when the investigation of arthropod-borne viruses provided an important rationale for the development of mosquito cell culture [56,57,58]. Dwight Lynn, formerly at the United States Department of Agriculture, Agricultural Research facility in Beltsville, Maryland, has described step-by step procedures for the development of insect cell lines [59,60,61]. Note that fetal bovine serum (often heat-treated to inactivate complement, a cytolytic component of mammalian innate immunity), and/or commercially available hydrolysates are sometimes used to supplement otherwise chemically defined cell culture media. About 500 insect cell lines have been described [61], but much remains to be done. For example, the first cell lines from honeybees, now threatened by pesticides, nutritional deficiency and pathogens, have only recently been established [62,63,64]. 

A permanent cell line consists of the population of dividing cells that emerges from a “primary culture”, which likely included more than one cell type. The population of cells that can sustain repeated passage may take considerable time to emerge. During early passages, population doubling is slow and successful subculture requires relatively low dilution of cells. Populations of cultured cells vary in their properties and growth patterns, which can evolve over time as continued maintenance in vitro “selects” for faster-growing cells. A few cell lines, such as the *Ae. albopictus* C7-10 cells used in the author’s lab, are a genetically homogeneous clonal population generated from an individual cell. This line, and its derivatives, can be grown in a modified vertebrate cell culture medium [65] and have been used to establish, with insect cells, the suite of protocols loosely called somatic cell genetics [54,55].

Differentiated insect tissues and organs may contain polytene or polyploid cells, which are unlikely to revert to a mitotic cycle [66], and cell lines are most easily established from embryonic or ovarian tissues or from imaginal discs. Although it is desirable to generate cell lines from specific tissues, robust methods for the rigorous correlation of a dividing cell population with its tissue of origin remain to be developed, and the presence and potential overgrowth of mitotic hemocytes in primary cultures remains a concern. Monoclonal antibodies [67] and staining properties [68] that have been used to distinguish among hemocyte populations have not been extended as a routine tool for correlating cell lines with tissues of origin.

Although large-scale proteomics has become an increasingly popular tool for identifying and exploring potential functions of novel proteins [69,70], the use of radiolabeled precursors combined with metabolic inhibitors remains an important tool for the initial identification of specific molecules that participate in molecular processes in insect cells and organs. Over the years, substantial improvements in the specific activity of isotopes have been achieved, and defined culture media, often commercially available, simplify experimental design. Radiolabeling is typically less efficient in media that contain undefined components such as yeast extract, lactalbumin hydrolysate or tryptose broth, but these supplements, which may be essential for long-term growth, may not be strictly necessary for short-term culture. Isotope incorporation is most effective when the concentrations of the corresponding non-labeled precursor are reduced or eliminated. For example, methionine-free medium is often used to label proteins with the relatively inexpensive ^35^[S]methionine/cysteine label derived from an *Escherichia coli* hydrolysate and available commercially. This approach, combined with Western blotting and mass spectrometry, suggested that host cell proteasome activity provides amino acids to *Wolbachia* [71]. There is a high likelihood that new soluble factors that mediate interactions among insect tissues [72,73] will eventually be chemically identified, particularly when positive interactions are investigated using newer proteomics technologies. Likewise, molecular approaches have provided important insights into expression of transfected genes in cultured mosquito cells [74], establishing the identity of closely-related cell lines [75] and exploring cellular responses to insect hormones [76,77,78].

## 4. Viruses and Microorganisms

In insects, the vertical transmission of mobile genetic elements, viruses and bacteria occurs through the egg, and it is not surprising that insect cell lines sometimes host endogenous viruses, such as the cell fusing agent (CFA), which does not cause cytopathology in host cells under routine culture conditions. CFA was identified when *Ae. aegypti* cells were inadvertently mixed with *Ae. albopictus* cells, which responded to the virus by undergoing fusion to generate a syncytium, followed by recovery of cells with a normal appearance [79]. In nature, most *Ae. aegypti* mosquito populations are infected with CFA, now known to be an insect-specific flavivirus [80] with possible effects on dissemination of arboviruses such as dengue [81,82]. Many lepidopteran cell lines are susceptible to baculoviruses, which have been used to propagate these insect-specific viruses for use as biocontrol agents, and more recently have been developed for expressing glycosylated mammalian proteins engineered into baculovirus-derived expression vectors [83]. Obligate intracellular bacteria can also be present in cultured cells, and of particular relevance to my own work was the recovery of a mosquito cell line infected with *w*AlbB, one of two strains of *Wolbachia* that occur in *Ae. albopictus* mosquitoes [37].

*Wolbachia* was described nearly a century ago as a *Rickettsia*-like bacterium in the ovaries of *Culex pipiens* mosquitoes [84,85]. Using antibiotics to eliminate *Wolbachia*, Yen and Barr showed that the bacterium, now known as *Wolbachia pipientis*, was the agent responsible for cytoplasmic incompatibility (CI) [86,87], a reproductive distortion resulting in failure of fertilized eggs to hatch [88,89,90,91]. Briefly, as an obligate intracellular bacterium, *Wolbachia* is propagated by maternal transmission in the egg cytoplasm. In a mixed population of infected and uninfected individuals, eggs of infected females hatch regardless of the infection status of the male, while eggs of uninfected females hatch only when fertilized by sperm from an uninfected male. The reproductive advantage of infected females allows *Wolbachia* to function as a gene drive agent that can be harnessed to replace vector with non-vector populations. As early as the 1960s, CI was recognized as a potential tool for population suppression and successfully tested in the field with a mosquito vector of filariasis [92,93]. 

Although roughly half of all insect species host *Wolbachia* [94], most natural populations of the malaria vector, *Anopheles gambiae* Say (Diptera, Anophelinae) and the yellow fever/dengue/Zika vector, *Ae. aegypti*, are not infected. Remarkably, when *Wolbachia* are artificially introduced into these species by microinjection of embryos, the surviving adults are infected, and exhibit generalized antipathogen effects that reduce the transmission of mosquito-borne diseases by an unknown mechanism independent of gene drive [95,96,97,98]. Potential applications of *Wolbachia*-based control strategies, including gene drive, pathogen resistance and prospects for eventual genetic manipulation provide a rationale for basic research on this intracellular bacterium, which is closely related to arthropod-borne pathogens of humans in the genera *Anaplasma*, *Ehrlichia* and *Rickettsia.* Aside from their occurrence in arthropods, mainly as reproductive parasites, *Wolbachia* also occur in filarial worms as essential symbionts.

## 5. Wolbachia in Cultured Cells

Since *Wolbachia* have streamlined genomes that lack some of the genes essential for growth as free-living bacteria, they cannot be cultured in liquid broth nor are they amenable to standard microbiological manipulations. A few cultured cell lines maintain a persistent *Wolbachia* infection derived directly from the source material. These include the *Ae. albopictus* Aa23 cell line that contains *w*AlbB [37], and a *Drosophila melanogaster* cell line, JW-18, from flies infected with a virulent strain of *Wolbachia* called *w*Mel-popcorn. Since *Wolbachia* from filarial worms have not been adapted to cell lines, the JW-18 infection has been used to screen for anti-filarial drugs [99]. A small number of *Wolbachia* strains have been introduced into various insect cell lines, including the Aa23T line, from which the endogenous *Wolbachia* were removed using tetracycline [100].

*Wolbachia* infections that have been maintained in cultured insect cells are from the most abundant pandemic A and B supergroups. Mosquito or *Drosophila* cell lines are the most typical hosts, but infections in these cells can be transferred to other host cells [100]. In the author’s lab, strain *w*Stri from the planthopper *Laodelphax striatellus* (Fallén); (Hemiptera, Delphacidae) [101] maintains a particularly robust infection in *Ae. albopictus* C7-10 cells [102], and the infected line, called C/*w*Stri1, has been used to optimize in vitro production of infectious *Wolbachia* [103] with the long-term goals of developing tools for its genetic manipulation and understanding the requirements for in vitro growth that may enable more recalcitrant *Wolbachia* strains to be adapted to cultured cells. In particular, the robust levels of *Wolbachia* in C/*w*Stri1 cells facilitated the development of a protocol for quantifying *Wolbachia* by flow cytometry [104]. 

An important goal of my ongoing research is to recover *Wolbachia* that remain capable of initiating a new infection after isolation and purification. In other words, I seek to complement viability tests based on cytological staining with verification that the live bacterium is infectious and capable of replication in a naive host cell, or after microinjection into an egg. A complementary goal is to identify properties of the host cell that minimize the size of an inoculum needed to establish a robust infection. Optimizing these parameters will facilitate eventual genetic manipulation of *Wolbachia*. Surprisingly, the effects of ecdysone on *Wolbachia* led to the observation that non-mitotic host cells that remain metabolically active support high yields of *w*Stri (Fallon submitted). 

Aside from *w*Stri, it is of interest to develop in vitro protocols for the *Wolbachia* strains that occur in mosquitoes. *Wolbachia*-induced cytoplasmic incompatibility is well-understood [90,91,105,106,107] in *Culex pipiens* mosquitoes, but it has been difficult to adapt *w*Pip to long-term growth in a cell line [108]. In our hands, the mosquito-associated strain *w*AlbB in Aa23 cells [37] grows slowly, requires high levels of serum and produces low yields of infectious *Wolbachia*, relative to *w*Stri [109]. Since *w*AlbB has successfully been introduced into *Ae. aegypti* by embryo microinjection [110], it is of interest to note that improved yields with mitotically inactivated host cells with *w*Stri extend to *w*AlbB, as described below.

Particularly informative has been the observation that the proliferation of *w*Stri increases when newly infected C7-10 cells reach the stationary phase of the cell cycle. Based on this observation, I tested whether ecdysone, which mediates an arrest in the G1 phase of the cycle in C7-10 cells [35], affects *Wolbachia* yields from C/*w*Stri1 cells. Ecdysone increases *Wolbachia* recovery per cell by about 10-fold, despite the overall inhibition of cell proliferation (Fallon, submitted). In contrast to C7-10 cells, Aa23 cells are not a clonal population, and little has been done to develop somatic cell genetics either with the *w*AlbB infected cells, or with tetracycline-cured derivatives. Although ecdysone has not been systematically studied in the context of the Aa23 cell cycle, these cells resemble C7-10 cells insofar as ecdysone increases *Wolbachia* production, particularly when near-confluent monolayers are used as the starting material and sufficient time has elapsed, presumably because Aa23 cells grow slowly (Figure 2). 

As background for interpretation of Figure 2, note that events detected by flow cytometry represent DNA fluorescence after staining with propidium iodide [104]. The *Wolbachia,* with smaller genomes (note the X-axis), are represented by the peak labeled WP, best shown in Figure 2d. Fluorescence from the larger host cell nuclei is represented by the set of peaks at the far right, reflecting the G1 (diploid) and G2 (tetraploid) phases of the host cell cycle, followed by a low frequency of doublets and/or tetraploid G1 cells extending to the extreme right. 

Figure 2 shows Aa23 cells maintained without ecdysone (panels (a) and (b) at left), or with 2 × 10^−6^ M ecdysone (panels (c) and (d) at right). During the first 5 days after plating the profiles in (a) and (c) show little difference other than a slight shift towards G1 in the treated cells (c). In untreated plates, cell (CP) and *Wolbachia* (WP) profiles change little between days 5 and 12, consistent with the slow growth of Aa23 cells and relatively low yields of *Wolbachia* noted previously [109]. With ecdysone treatment *Wolbachia* increases, cells numbers drop and in the nuclei that remain, G1 exceeds G2 on day 12 (Figure 2; compare (c) and (d)). The nature of the slight shoulder at the right of the main WP in Figure 2d is unknown, but is consistently indicative of a robust infection with both *w*Stri and *w*AlbB.

The welcome increase in *Wolbachia* recovery on a per cell basis after ecdysone treatment was somewhat counterproductive in terms of absolute *Wolbachia* yields because the hormone arrests cell growth and therefore limits the number of cells from which *Wolbachia* can be harvested. Insofar as *Wolbachia* replicate in stationary phase cells, and yield is enhanced when host cell cycling is inhibited by ecdysone, it was of interest to explore whether *Wolbachia* infect and replicate in mitotically inactivated cells similar to feeder layers used to support embryonic stem cells from mammalian sources. Success with this approach would enable the use of high numbers of host cells to maximize *Wolbachia* yields.

Mitotic inactivation can be achieved by treating uninfected C7-10 cells with mitomycin C, which crosslinks DNA and arrests mitosis while allowing cells to remain metabolically active [111]. Inactivated cells, when infected with purified *w*Stri, support increased yields of infectious *Wolbachia* comparable to those achieved with ecdysone, and reduce the multiplicity of infection required for a robust yield to as few as a single *w*Stri bacterium per cell (Fallon, submitted). In contrast, with naive exponentially growing C7-10 cells as hosts, approximating 80 *w*Stri per cell are needed for robust recovery of progeny *Wolbachia* [103]. It appears that actively growing cells limit *Wolbachia* replication by an as of yet unknown mechanism that lessens when the host cell is mitotically quiescent.

Ongoing experiments show an approximately 25-fold improvement in *w*AlbB yield when Aa23 cells are maintained on confluent monolayers of mitomycin C-treated C7-10 cells. Figure 3 shows a dot plot (a) and histogram (b) of flow cytometry data from growing Aa23 cells. Unlike growing C7-10 cells, where few events occur in Q1 [104], Aa23 cells show abundant events in the Q1 quadrant of a scatter plot. This unidentified material skews the symmetry of the WP shown in Figure 3b, and overestimates *Wolbachia* abundance (note the open double arrow connecting panels (a) and (b)). With lower numbers of Aa23 cells, such as the ecdysone experiment shown in Figure 2d, Q1 events are less problematic, and the WP is more symmetric. Moreover, Q1 events also occur with C/*w*Stri1 cells late in the *Wolbachia* replication cycle, after recovery of infectious bacteria drops [103,104]. It is not yet clear whether these events represent the breakdown of mitochondria and/or degraded DNA correlated with apoptosis of the infected cells. However, the appearance of this material in Aa23 cells earlier in the infection cycle and at lower cell densities is consistent with lower yields of *Wolbachia* and may be indicative of a more virulent infection. Preliminary studies with C/*w*Stri1 cells suggest that Q1 events do not result from lytic expression of a *Wolbachia* prophage.

To compensate for Q1 events, *Wolbachia* abundance in Aa23 cells can be estimated based on events gated in polygon T, rather than events in the WP (Figure 3, compare panels (a) and (b)). Figure 3c,d represent growth of a second population of Aa23 cells distributed onto 100 mm plates in the absence (c) and presence (d) of a confluent monolayer of mitomycin C-treated, mitotically inactivated C7-10 cells. After two weeks, overall yields of *w*AlbB with feeder layers were increased 25-fold (compare material in the T gate in panels (c) and (d)). Note that the difference in cell numbers in Q2 panels in the upper right quadrant is generated by nuclei of the mitotically inactivated C7-10 cells.

The microscopic appearance of the *w*AlbB infection arising when Aa23 cells have been co-cultured with mitotically inactive C7-10 cells for two weeks (Figure 3d) is shown in Figure 4a. At this magnification, individual *Wolbachia* are represented by the speckling of tiny green dots best noted in the oval area to the left of the white arrow; see also [103]. The distribution of *Wolbachia* fluorescence shows that *w*AlbB readily invades and proliferates in the mitomycin C-treated C7-10 cells, which often expanded into syncytia-like structures approaching a diameter of 50 microns and reminiscent of a CFA infection [79]. Inoculated Aa23 cells with lower levels of infection have a diameter of approximately 10 microns and are shown in the rectangle, with the shorter arrow indicating a live cell with a green nucleus, and few if any *Wolbachia*, and the longer arrow showing a dead cell with a red nucleus. The extent to which the feeder layer improves replication and release of *Wolbachia* from Aa23 cells, relative to the proportion of *w*AlbB replicating in the C7-10 feeder cells, remains to be determined, as does the possibility that fusion reflects an unidentified endogenous virus.

After the purification by filtration and centrifugation as detailed elsewhere [103], final yields of infectious, freely floating *w*AlbB showed a typical Q3 FACS profile (Figure 4b) that lacked the unknown material present in the Q1 quadrant of the starting inoculum shown in Figure (3a,c,d). Final yields of *w*AlbB defined previously as the more infectious freely floating *Wolbachia* (FFW) [103] were as high as 10^9^ bacteria per 100 mm plate of mitomycin C-treated C7-10 cells, and were comparable to levels obtained with *w*Stri. 

## 6. Conclusions

I anticipate that the continued improvement of *Wolbachia* recovery from mitotically inactivated cells will provide a procedure for culturing recalcitrant strains and for producing sufficient *Wolbachia* for mutagenesis and eventual transformation, in which the recovery of manipulated cells is expected to be fewer than one per million. Applying these approaches to an obligate intracellular bacterium that occurs in many insects is of basic interest, and may have important practical applications in microbiology and in pest control. Mitotic inactivation seems to reduce negative host cell responses to *Wolbachia*, while providing nutritional reserves on which *Wolbachia* depend for replication. Further exploration of nutritional supplementation of mitotically inactivated cultures and application of in vitro technologies that have been used in mammalian stem cells may eventually contribute to new approaches for the culture of insect symbionts and to environmentally compatible strategies for *Wolbachia*-based control of medical and agricultural pests. Finally, in future studies it will be of interest to explore how hormonal cues that support initiation and progression of the vitellogenic cycle influence *Wolbachia* replication and transmission to subsequent generations via infected eggs.

## Figures and Tables

**Figure 1 insects-13-00756-f001:**
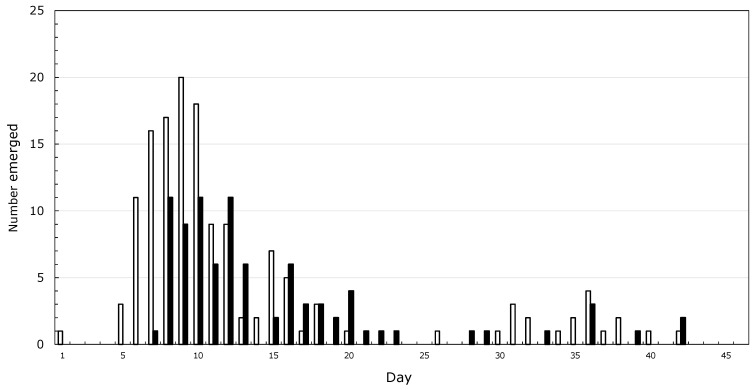
Emergence of *Hyalophora cecropia* moths from 1 June through 15 July 1965 in Pawcatuck, CT, USA. Open bars, males; closed bars, females.

**Figure 2 insects-13-00756-f002:**
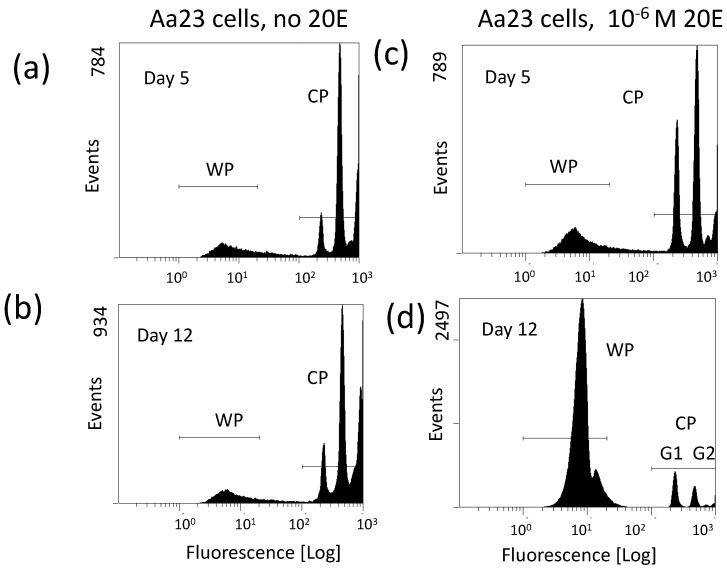
Ecdysone increases *Wolbachia* abundance in Aa23 cells. Panels (**a**,**b**) show profiles of Aa23 cells in the absence of ecdysone (20E) on day 5 and day 12 after plating cells at near confluence. Panels (**c**,**d**) show replicate plates with 2 × 10^−6^ M ecdysone added on day 0. Note that fluorescence corresponding to *Wolbachia* is represented by WP; fluorescence due to host cell nuclei is shifted to the far right (CP). Events on the Y-axis in panels (**a**,**c**) are similar, suggesting little effect of ecdysone during the first 5 days of exposure. On day 12, the WP has substantially increased in panel (**d**) relative to (**b**), reflecting improved yield of *Wolbachia*, and the CP has dropped, reflecting the negative effect of ecdysone on cell proliferation over time. Note that the CP profile in (**d**) shows a higher proportion of cells in G1 than in G2, as is the case with C7-10 cells [35]. This experiment was replicated with three independent biological samples.

**Figure 3 insects-13-00756-f003:**
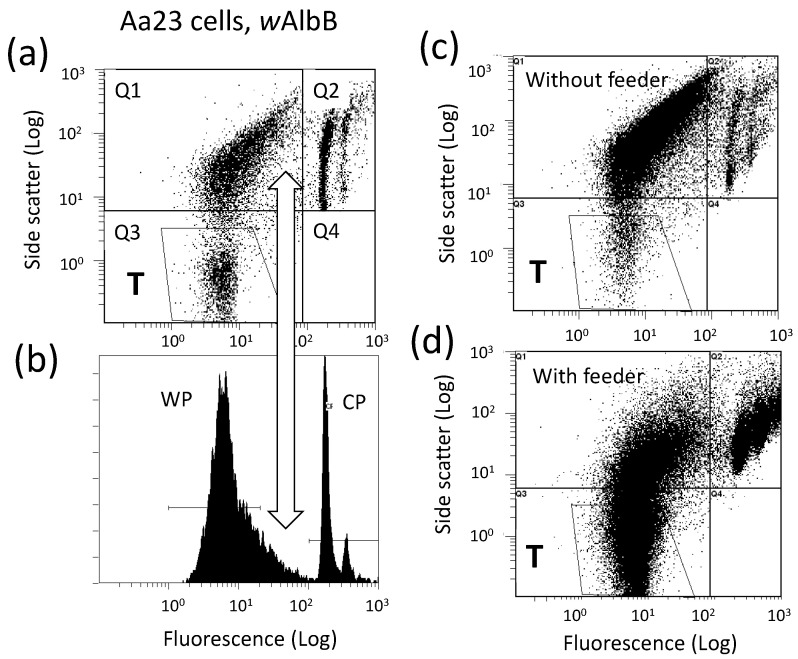
Cytometric analysis of crude *Wolbachia* preparations from Aa23 cells directly resuspended in culture medium. Panel (**a**), scatter plot showing events in Q1 that interfere with quantitation of the WP in the corresponding histogram shown in Panel (**b**). The vertical white open arrow indicates corresponding positions in the two panels. WP, *Wolbachia* peak; CP, cell peak. Panel (**c**), an independent preparation of Aa23 cells assayed after two weeks in the absence of a C7-10 feeder layer; Panel (**d**), a replicate Aa23 inoculum added to a confluent monolayer of mitomycin C-treated C7-10 cells. Events in polygon T, which excludes Q1 events (shown in (**a**,**c**,**d**)) were used to estimate relative *Wolbachia* abundance.

**Figure 4 insects-13-00756-f004:**
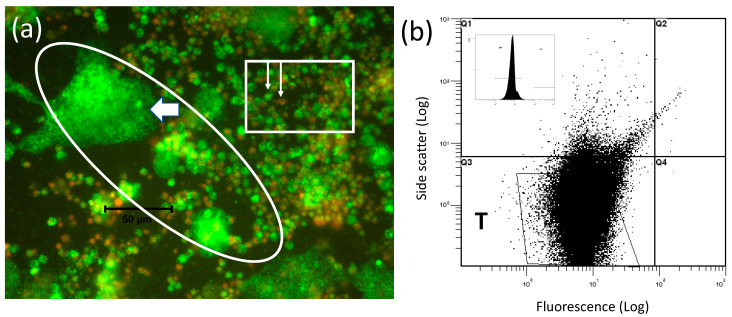
*w*AlbB harvested from C7-10 cells treated with mitomycin C. Panel (**a**) shows an image of cells represented in Figure 3d after a two-week incubation with mitomycin C-treated feeder cells. Based on size (approximately 10-micron diameter) and appearance of nuclei, cells shown in the rectangle are Aa23 cells introduced as the inoculum. White arrows show nuclei of live (green) and dead (red) Aa23 cells. The oval region shows the aberrant morphology and high levels of *Wolbachia* (barely visible, small green dots such as those in the syncytium-like structure to the left of a white, leftward pointing arrow) that develop in the feeder layer. The black size marker that crosses the oval at left indicates 50 microns. Panel (**b**) shows the final *w*AlbB scatter plot after purification by filtration and centrifugation as described previously for C/*w*Stri1 cells [103]. The inset shows the WP profile.

## Data Availability

Not applicable.

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
