# Peer review of "From Mosquito Ovaries to Ecdysone; from Ecdysone to Wolbachia: One Woman’s Career in Insect Biology"

_insects, 2022, doi:10.3390/insects13080756_

Round 1

Reviewer 1 Report

These are my main comments on the manuscript (insects-1845179) entitled “From mosquito ovaries to ecdysone; from ecdysone to Wolbachia: One woman's career in insect biology”. The review summarizes studies involving the insect steroid hormone ecdysone and as affects the process of cell division in cultured mosquito cells, causing a cell cycle arrest that enhances growth of Wolbachia pipientis. Following moderate revisions should be incorporated in the manuscript prior to acceptance.

Ls.23, 122, 284, 358: in vitro should be in italic.

Ls.32, 56, 59, 66, etc.: For each scientific name species, provide the ID author, order and family taxa. Correct in all manuscript.

L.35: Keywords should be in alphabetic order. Also, keywords serve to widen the opportunity to be retrieved from a database. To put words that already are into title and abstracts makes KW not useful. Please choose terms that are neither in the title nor in abstract.

L.38: …called lady bugs, a…

L.42: … I had neighbors, and…

L.85: …Wyatt, I investigated…

L.85: Define “GMP”

L.96: …Eventually, I accepted…

L.104: Define “RNA”

L.112: … cells [37], I began…

L.146: in vivo and in vitro should be in italic.

L.151: …trigger the blood…

L.162: … diapause, and migration…

L.180: Define “USDA”

L.181: Define “MD”

Author Response

Reviewer 1

Comments and Suggestions for Authors

#1.   These are my main comments on the manuscript (insects-1845179) entitled “From mosquito ovaries to ecdysone; from ecdysone to Wolbachia: One woman's career in insect biology”. The review summarizes studies involving the insect steroid hormone ecdysone and as affects the process of cell division in cultured mosquito cells, causing a cell cycle arrest that enhances growth of Wolbachia pipientis. Following moderate revisions should be incorporated in the manuscript prior to acceptance.

Ls.23, 122, 284, 358: in vitro should be in italic.

Italics are added as suggested

Ls.32, 56, 59, 66, etc.: For each scientific name species, provide the ID author, order and family taxa. Correct in all manuscript.

ID author, order and family are provided at first mention, except in the simple summary.

L.35: Keywords should be in alphabetic order. Also, keywords serve to widen the opportunity to be retrieved from a database. To put words that already are into title and abstracts makes KW not useful. Please choose terms that are neither in the title nor in abstract.

Key words have been alphabetized and duplication of title words has been eliminated.

I have made all of the minor changes recommended by reviewer 1 below, with the exception that I have not spelled out the abbreviation for RNA, which will be familiar to readers of this journal.

L.38: …called lady bugs, a…

L.42: … I had neighbors, and…

L.85: …Wyatt, I investigated…

L.85: Define “GMP”

L.96: …Eventually, I accepted…

L.104: Define “RNA”

L.112: … cells [37], I began…

L.146: in vivo and in vitro should be in italic.

L.151: …trigger the blood…

L.162: … diapause, and migration…

L.180: Define “USDA”

L.181: Define “MD”

Reviewer 2 Report

This manuscript provides a very nice review on how a well-respected insect cell biologist develops her career in science. It is very pleasant reading and can inspire the young generation with interests in insect biology.  The readers will benefit from history of knowledge, in-depth understanding of Wolbachia-host interactions and its insightful discussion. In addition to reviewing the work in the field, there are several highly innovative, significant and interesting works described in the manuscript. Probably they are working progress, some evidence appears vague. It can better advance the field if additional information is provided as described below.   

“Infectious” Wolbachia is a very novel concept and could be a significant discovery in the field. The author is suggested to expand this topic and provide data to support this claim and explain explicitly how it is defined, what makes infectious Wolbachia different from other Wolbachia, and why Wolbachia has infectious and non-infectious status.

Author is suggested to provide either direct data or cite paper to show ecdysone treatment increases Wolbachia abundance in cells. Is there any in vivo evidence consistent with this in vitro observation? As ecdysone treatment causes broad physiological change in host, why author believes that it is its induced cell cycle arrest, rather than others, that increase Wolbachia growth.  

Wolbachia are represented by signals in polygon of Q3, which is essential evidence to interpret both figure 2 and 3 as described in main text. However, a control group, using Wolbachia-free cells, is missed. If the signals in Q3 present only in Wolbachia-infected cells but not in the control, one would be confident they are associated with Wolbachia.

Figure 4, it is unclear how author determines that the oval region represents C7-10 cells and cells in the square are Aa23 cells.   Please also point Wolbachia using a specific symbol.   

Line 254, it is population suppression, rather than population replacement, that CI was used as a tool for vector control in 1960’s, as supported by Laven’s work (1967).

Line 280, they should be A and B supergroups, not strains. (Werren, et al. 2008. Nature Reviews Microbiology).

Line 290, maybe one of best understood. CI mechanism is better, at least not less, understood for wMel than wPip.

Author Response

Response to Reviewer 2

#2.  Comments and Suggestions for Authors

This manuscript provides a very nice review on how a well-respected insect cell biologist develops her career in science. It is very pleasant reading and can inspire the young generation with interests in insect biology.  The readers will benefit from history of knowledge, in-depth understanding of Wolbachia-host interactions and its insightful discussion. In addition to reviewing the work in the field, there are several highly innovative, significant and interesting works described in the manuscript. Probably they are working progress, some evidence appears vague. It can better advance the field if additional information is provided as described below.   

Reviewer 2 suggested some important changes that substantially improve the manuscript, for which I am grateful.  These are mostly in Section 5 and include extensive revisions to Figures 2, 3 and 4 and accompanying text. 

“Infectious” Wolbachia is a very novel concept and could be a significant discovery in the field. The author is suggested to expand this topic and provide data to support this claim and explain explicitly how it is defined, what makes infectious Wolbachia different from other Wolbachia, and why Wolbachia has infectious and non-infectious status.

The revised text describes the distinction between live versus infectious Wolbachia (lines 295-302) that underlies current investigations. 

Author is suggested to provide either direct data or cite paper to show ecdysone treatment increases Wolbachia abundance in cells. Is there any in vivo evidence consistent with this in vitro observation? As ecdysone treatment causes broad physiological change in host, why author believes that it is its induced cell cycle arrest, rather than others, that increase Wolbachia growth.  

As requested, I have revised Fig. 2 to document the ecdysone response in Aa23 cells (lines 331-340).  In this figure, a nearly negative Wolbachia peak (corresponding to Q3 events) is intended to satisfy the request for a WP control.

Wolbachia are represented by signals in polygon of Q3, which is essential evidence to interpret both figure 2 and 3 as described in main text. However, a control group, using Wolbachia-free cells, is missed. If the signals in Q3 present only in Wolbachia-infected cells but not in the control, one would be confident they are associated with Wolbachia.

Original figures 2 and 3 have been merged to produce a new Figure 3.  The polygon T is more clearly explained, and reference is made to the original methods paper [104] for comparative flow cytometry details with C/wStri cells.  More detail is described for events in quadrant Q1 (lines 359-374).

Figure 4, it is unclear how author determines that the oval region represents C7-10 cells and cells in the square are Aa23 cells.   Please also point Wolbachia using a specific symbol.   

Figure 4 has been clarified with additional arrows, and panels reversed to better support the relation between Fig. 3 (crude) and Fig. 4 (purified) samples from a single population of Aa23 cells.

Line 254, it is population suppression, rather than population replacement, that CI was used as a tool for vector control in 1960’s, as supported by Laven’s work (1967).

Yes, it should read population suppression (line 259)

Line 280, they should be A and B supergroups, not strains. (Werren, et al. 2008. Nature Reviews Microbiology).

Yes, supergroups is correct (line 285) 

Line 290, maybe one of best understood. CI mechanism is better, at least not less, understood for wMel than wPip.

Yes, "one" of the best is correct (line 305)

Reviewer 3 Report

Overview

Overall, author Ann Fallon does a great job with this review on mosquito reproduction to Wolbachia. This reviewer greatly appreciated the history lesson of this well written review as it relates to the current state of the field regarding vector management/replacement.

The only issues I have are with the flow cytometry data, specifically dot plot labeling. While I do understand “Q1 – Q4” as these are standard output from flow cytometry programs, I am unclear the designation of “T” in Figure 2C. Why use T as to another character? Also, in the figure legend, “T” is not referenced.

Also, is the point of the flow data to suggest that slowly dividing cells make better hosts for Wolbachia? I would find it hard to believe that simply slowing division of cells is all that is need to make a suitable “home” for Wolbachia, or am I wrong?   

Author Response

#3.  Overall, author Ann Fallon does a great job with this review on mosquito reproduction to Wolbachia. This reviewer greatly appreciated the history lesson of this well written review as it relates to the current state of the field regarding vector management/replacement.

The only issues I have are with the flow cytometry data, specifically dot plot labeling. While I do understand “Q1 – Q4” as these are standard output from flow cytometry programs, I am unclear the designation of “T” in Figure 2C. Why use T as to another character? Also, in the figure legend, “T” is not referenced.

I have explained in more detail why non-dividing cells appear to be better hosts for Wolbachia (lines 342-358), and revisions requested by Reviewer 2 improve the description of these issues.  T is described in the legend to revised Figure 3.

 Also, is the point of the flow data to suggest that slowly dividing cells make better hosts for Wolbachia? I would find it hard to believe that simply slowing division of cells is all that is need to make a suitable “home” for Wolbachia, or am I wrong?   

It's more complicated than slowing division, as simply using serum-free medium doesn't work, nor do other cell cycle inhibitors that arrest at different stages of the cycle.  This is a complex issue best left for more experimentation and a future manuscript.  In the conclusions, I note that these findings need to be integrated into an understanding of reproductive physiology (lines 411-418).